# Er-Doped BiVO_4_/BiFeO_3_ Nanocomposites Synthesized via Sonochemical Process and Their Piezo-Photocatalytic Application

**DOI:** 10.3390/nano14110954

**Published:** 2024-05-29

**Authors:** Thanaphon Kansaard, Maneerat Songpanit, Russameeruk Noonuruk, Chakkaphan Wattanawikkam, Wanichaya Mekprasart, Kanokthip Boonyarattanakalin, Chalicheemalapalli Kulala Jayasankar, Wisanu Pecharapa

**Affiliations:** 1College of Materials Innovation and Technology, King Mongkut’s Institute of Technology Ladkrabang, Bangkok 10520, Thailand; tkansaard@gmail.com (T.K.); 66116007@kmitl.ac.th (M.S.); wanichaya.me@kmitl.ac.th (W.M.); kanokthip.bo@kmitl.ac.th (K.B.); 2Faculty of Science and Technology, Rajamangala University of Technology Thanyaburi, Khlong Luang 12110, Thailand; russameruk@gmail.com (R.N.); chakkaphan_w@rmutt.ac.th (C.W.); 3Department of Physics, Sri Venkateswara University, Tirupati 157520, India; ckjaya@yahoo.com

**Keywords:** piezo-photocatalyst, BiVO_4_/BiFeO_3_, nanocomposites, sonochemical process

## Abstract

In this work, Er-doped BiVO_4_/BiFeO_3_ composites are prepared using the sonochemical process with a difference of rare earth loading compositions. The crystallinity and chemical and morphological structure of as-synthesized samples were investigated via X-ray diffraction, Raman scattering, and electron microscopy, respectively. The diffuse reflectance technique was used to extract the optical property and calculate the optical band gap of the composite sample. The piezo-photocatalytic performance was evaluated according to the decomposition of a Rhodamine B organic compound. The decomposition of the organic compound was achieved under ultrasonic bath irradiation combined with light exposure. The Er-doped BiVO_4_/BiFeO_3_ composite heterojunction material exhibited significant enhancement of the piezo-photocatalytic activity under both ultrasonic and light irradiation due to the improvement in charge generation and separation. The result indicates that Er dopant strongly affects the phase transformation, change in morphology, and alternation in optical band gap of the BiVO_4_ matrix. The incorporation of BiFeO_3_ in the composite form with BiVO_4_ doped with 1%Er can improve the photocatalytic performance of BiVO_4_ via piezo-induced charge separation and charge recombination retardment.

## 1. Introduction

For decades, photocatalysts based on metal oxide semiconducting compounds, including titanium dioxide (TiO_2_) [1], zinc oxide (ZnO) [2], cerium oxide (Ce_2_O_3_) [3], and bismuth oxide (Bi_2_O_3_) [4], have been extensively explored and practically utilized for pollutant treatment. However, these catalyst compounds have considerable drawbacks, including their wide band gap energy, which requires high photon energy of the ultraviolet spectrum to activate their functionality. The bismuth vanadate (BiVO_4_) compound, with a lower optical band gap aligning in the visible region, has gained attraction as an effective visible-driven photocatalyst for practical usages due to its significant advantages, including non-toxic metal oxide, high stability, good photocatalytic activity, low recombination rate of photogenerated charges, and capability of visible activation. To further enhance the performance of metal oxide-based photocatalysts, doping with proper elements incorporated into the host material is considered as an effective way to adjust its optical band gap and retard the electron–hole pair recombination. Interestingly, it has been reported that rare earth (RE) dopants in BiVO_4_ can lead to the substitution of RE ions at the Bi-site, resulting in not only significant suppression of electron–hole recombination due to the specific electronic transition property, but also good crystal structure stabilization [5]. In addition, due to strong photon absorption and emission of rare earth ions such as Yb^3+^, Er^3+^, and Y^3+^, it is perceived that the incorporation of Er ions into BiVO_4_ could not only adjust the morphology, but also increase the optical absorption ability of BiVO_4_, resulting in an enhancement in its photocatalytic performance [6]. Moreover, it has been recently reported that the catalytic performance of the catalysts can be further improved by assistance with piezoelectric materials via the piezo-photocatalysis process. Under light illumination and ultrasonic irradiation, photo-induced and thermal-activated charges, including electrons and holes, can be generated. Simultaneously, the polarization electric field induced in a piezo-photocatalyst could lead to effective migration and separation of these excited charges [7]. Moreover, heterojunction between two semiconductors in the form of a composite is considered as an effective strategy to force charge separation due to different energy band potentials and the induced interfacial electric field. It is expected that the formation of heterojunction and the modification of the piezo-induced electric field across the catalysts could effectively improve the catalytic performance. Among various piezoelectric compounds, BiFeO_3_, with its rhombohedral structure and appropriate optical band gap of ~2.2 eV, is considered as potential compound regarding its exceptional piezoelectricity, pyroelectricity, ferroelectricity, and good visible-driven photocatalyst [8]. Furthermore, BiFeO_3_ is recognized as an important piezo-electric material owing to its non-centrosymmetric structure with space group R3c and point group 3m. However, it is somewhat difficult to synthesize phase-pure BiFeO_3_, since it has a narrow temperature range of phase stabilization [9]. Until now, several BiFeO_3_-based heterojunction structures, including In_2_O_3_/BiFeO_3_ [8], ZnIn_2_S_4_/BiFeO_3_ [10], and Carbon-dots/BiFeO_3_ [11] have been constructed. For the synthesis process of metal oxide-based compounds, the sonochemical process can be considered as an effective route due to the simplicity of its equipment, the rapid particle formation process assisted by intense ultrasonic energy, and the ease of doping during the process [12]. However, the development of heterojunction between BiVO_4_-based photocatalyst and BiFeO_3_ and the synthesis of the functional piezo-photocatalyst composite via a sonochemical method are still rarely reported.

In this work, a facile sonochemical process was employed to synthesize Er-doped BiVO_4_/BiFeO_3_ composite. Relevant characterization methods, including XRD, DRS Raman, and TEM, were carried on as-prepared samples. The piezo-photocatalytic performance of all samples was conducted, and the influence of the Er dopant and incorporation of BiFeO_3_ on the photocatalytic properties of BiVO_4_ was investigated.

## 2. Materials and Methods

### 2.1. BiFeO_3_ Preparation

A pristine BiFeO_3_ (BFO) sample was prepared through a sonochemical and thermal treatment process. Initially, bismuth nitrate pentahydrate (Bi(NO_3_)_3_·5H_2_O) served as the Bi precursor, and iron nitrate nonahydrate (Fe(NO_3_)_3_·9H_2_O), designated as Fe precursor, were separately dissolved in deionized (DI) water, with a 1:1 mole ratio between Bi and Fe. After continuous stirring at room temperature for 10 min, both solutions were mixed, and the pH was adjusted to 9. The mixture was then stirred for an additional 10 min. Subsequently, the mixed solution was transferred to a sonochemical vessel and exposed to intense ultrasonic sound (750 W, 20 kHz) for 30 min. The product was separated, washed with DI water until it reached a neutral pH, and then dried at 100 °C overnight to obtain a brownish powder. Finally, the as-prepared powder was calcined at 500 °C for 2 h to produce the BFO powder.

### 2.2. BiVO_4_/BiFeO_3_ Composite Preparation

The BiVO_4_/BiFeO_3_ composite (BVO/BFO) sample was prepared with the sonochemical method. First, 0.1 moles of bismuth nitrate pentahydrate and vanadium metavanadate were assigned as Bi and V starting materials, and were dissolved in DI water in separated containers. Then, the Bi and V starting solutions were mixed, and the pH was adjusted to 7. The prepared BFO powder, with a designated BVO:BFO ratio of 1:1, was loaded into the solution under continuous stirring for 10 min. After that, the mixed solution was transferred to a sonochemical reactor under strong ultrasonic irradiation operated at 20 kHz with a power of 750 W for 30 min. The precipitated composite was washed to neutral and dried overnight to obtain a well-formed powder.

### 2.3. Erbium Doped BiVO_4_/BiFeO_3_ Composite Powder Preparation

An erbium (Er)-doped BVO/BFO composite was prepared with different Erbium loading contents using the sonochemical process. The experimental steps were the same as those used in the process of preparing the BVO/BFO composite, with the addition of dissolved erbium nitrate pentahydrate (Er(NO_3_)_3_·5H_2_O) assigned as the Er starting material. The Er loading concentrations in the composite were designated at 1%, 3%, and 5% mole.

### 2.4. Composite Film Preparation

To prepare the composite films for impedance evaluation, 50 mg of the composite was initially dispersed in 1 mL of ethanol solvent using an ultrasonic bath for 10 min. Subsequently, 1 mL of polyvinyl alcohol solution with a concentration of 5 wt% was added to the composite solution and stirred at room temperature for 10 min to obtain a homogenous, muddy solution. Following this, 100 µL of the solution was coated onto cleaned ITO substrates using a dropping casting technique, resulting in films with dimensions of 1.5 × 1.5 cm^2^. The coated substrates were then dried overnight at 100 °C and subsequently calcined at 300 °C for 5 h to remove the solvent and binder polymer, respectively.

### 2.5. Characterizations

The material characteristics were investigated using different techniques. The crystalline structure of the synthesized powder was observed via the X-ray diffraction (XRD) technique in the position range of 2θ= 20°−80° using a Rigaku SmartLab diffractometer (Smart lab, Rigaku Corp., Tokyo, Japan). The optical property was evaluated via diffuse reflectance spectroscopy (DRS) using a HITACHI UH4150 spectrophotometer, (Hitachi High-Tech Corp., Ibaraki, Japan). The chemical vibration characteristic was revealed with the Raman scattering technique using a Raman DRX smart Raman spectrophotometer (Thermo Fisher Scientific, Waltham, MA, USA). Surface properties such as morphology and specific surface area were observed using a transmission electron microscope (TEM) and the Brunauer, Emmett, and Teller (BET) technique via JEOL JEM-2100Plus microscopy (JEM-2100plus, JEOL Ltd., Tokyo, Japan) and a Quantachrome Autosorb iQ-C-XR-XR-XR machine (Anton Paar ltd., Albans, Hertfordshire, UK) respectively. Piezo-photocatalytic activity was performed with the decomposition of Rhodamine B (RhB) using an organic compound model under Xe-Lamp (300 W) (Sciencetech Inc., London, ON, Canada) combined with ultrasonic irradiation operated with a power of 200 W. Electrochemical impedance spectroscopy (EIS) was employed to assess the resistance to electron transfer during photocatalytic activity. EIS measurements of BVO/BFO and 1% Er-doped BVO/BFO composites were conducted using a three-electrode setup (Autolab PGSTAT302N, Metrohm AG, Herisau, Switzerland) with Ag/AgCl as a counter-electrode, Pt as a reference electrode, and the fabricated composite film as a working electrode.

## 3. Results and Discussion

### 3.1. Crystalline Structure

Figure 1 displays the X-ray diffraction (XRD) patterns, which reveal the unique crystalline structures of both as-prepared BFO and composite specimens. For the as-prepared BFO sample, the dominant feature is indexed by a prominent peak at 28.1°, indicating a cubic structure of bismuth oxide (Bi_2_O_3_), as identified by reference to JCPDS No. 00-042-0201. However, following the calcination of the as-prepared sample at 500 °C for 2 h, the XRD result reveals robust diffraction peaks at 22.4°, 32.0°, 39.6°, 45.8°, 51.4°, and 57.1°, which correspond to specific crystallographic planes of (012), (110), (202), (024), (116), and (214), respectively. This pattern indicates the presence of a trigonal structure, characteristic of bismuth ferrite (BiFeO_3_), which is well indexed to the reference pattern JCPDS No. 01-075-9475. In the case of the bismuth vanadate/bismuth ferrite (BVO/BFO) composite sample, the XRD results exhibit multiple diffraction patterns. These patterns suggest a combination of the BiFeO_3_ and bismuth vanadate (BiVO_4_) monoclinic crystal structures. Notably, the XRD peaks are situated at 18.9° and 28.9°, corresponding to the crystallographic planes (101) and (−112), nicely matching with the reference pattern JCPDS No. 01-083-1699. The proposed mechanism for BiFeO_3_ formation in this synthesis occurs through a two-stage process: For the first stage, bismuth nitrate pentahydrate (Bi(NO_3_)_3_·5H_2_O) and iron nitrate nonahydrate (Fe(NO_3_)_3_·9H_2_O) are dissociated in deionized water to form Bi^3^⁺, Fe^3^⁺, NO_3_^−^, and H_2_O at room temperature. Then, at pH 9, incorporated with ultrasonic sound, some Bi^3^⁺ and Fe^3^⁺ ions react with the surrounding water molecules, undergoing hydrolysis. This leads to the formation of partially hydroxylated Bi and Fe species, such as BiO(OH) and FeO(OH). Next, these partially hydroxylated Bi and Fe species might interact and form mixed-metal precursors with structures like Bi(OH)_x_Fe(OH)_y_ (where x and y represent the number of hydroxyl groups attached). However, the exact structure and extent of this interaction depend on various factors and might be difficult to predict precisely.

In the second stage, at 500 °C, thermal decomposition, oxidation, and crystallization occur. During calcination, the mixed-metal precursors undergo dehydration, releasing water molecules. Additionally, nitrate ions (NO_3_^−^) decompose through a combination of thermal and potentially oxidative mechanisms. The specific products formed (NO, NO_2_, and O_2_) depend on temperature and the presence of metal oxides (MO). Then, the remaining Bi and Fe cations react with oxygen to form Bi^3^⁺ and Fe^3^⁺ oxides (likely Bi_2_O_3_ and Fe_2_O_3_). At this elevated temperature, Bi_2_O_3_ and Fe_2_O_3_ can react in the solid state to form the perovskite structure of BiFeO_3_.
(1)$$\mathrm{Bi}{\left({\mathrm{NO}}_{3\;}\right)}_{3}•5H_{2}O_{\left(s\right)}\to {\mathrm{Bi}}_{\left(\mathrm{aq}\right)}^{3+}+3{\mathrm{NO}}_{3\;\left(\mathrm{aq}\right)}^{-}+5H_{2}O_{\left(l\right)}$$
(2)$$\mathrm{Fe}{\left({\mathrm{NO}}_{3}\right)}_{3}•2H_{2}O_{\left(s\right)}\to {\mathrm{Fe}}_{\left(\mathrm{aq}\right)}^{3+}+3{\mathrm{NO}}_{3\;\left(\mathrm{aq}\right)}^{-}+9H_{2}O_{\left(l\right)}$$
(3)$${\mathrm{Bi}}_{\left(\mathrm{aq}\right)}^{3+}+2H_{2}O_{\left(l\right)}\to \mathrm{BiO}{\left(\mathrm{OH}\right)}_{\left(\mathrm{aq}\right)}+3H_{\left(\mathrm{aq}\right)}^{+}$$
(4)$${\mathrm{Fe}}_{\left(\mathrm{aq}\right)}^{3+}+2H_{2}O_{\left(l\right)}\to \mathrm{FeO}{\left(\mathrm{OH}\right)}_{\left(\mathrm{aq}\right)}+3H_{\left(\mathrm{aq}\right)}^{+}$$
(5)$$\mathrm{BiO}\left(\mathrm{OH}\right)+\mathrm{FeO}\left(\mathrm{OH}\right)\to \mathrm{Bi}{\left(\mathrm{OH}\right)}_{x}\mathrm{Fe}{\left(\mathrm{OH}\right)}_{y}$$
(6)$$\mathrm{Bi}{\left(\mathrm{OH}\right)}_{x}\mathrm{Fe}{\left(\mathrm{OH}\right)}_{y\left(s\right)}\to {\mathrm{BiO}}_{x}{\mathrm{FeO}}_{y\;\left(\mathrm{aq}\right)}+\frac{\left(x+y\right)}{2}H_{2}O_{\left(g\right)}$$

Thermal decomposition:(7)$${\mathrm{NO}}_{3\;\left(\mathrm{aq}\right)}^{-}\to {\mathrm{NO}}_{2\;\left(\mathrm{aq}\right)}^{-}+{\;O}_{\left(g\right)}$$
(8)$${\mathrm{NO}}_{2\;\left(\mathrm{aq}\right)}^{-}\to {\mathrm{NO}}_{2\;\left(g\right)}+{\;O}_{\left(g\right)}$$
or
(9)$$\left(2{\mathrm{NO}}_{2\;\left(g\right)}\to 2{\mathrm{NO}}_{\left(g\right)}+{\;O}_{2\;\left(g\right)}\right)$$

Oxidative decomposition:(10)$$2{\mathrm{NO}}_{3\;\left(\mathrm{aq}\right)}^{-}+3{\mathrm{MO}}_{\left(s\right)}\to {\mathrm{NO}}_{2\;\left(g\right)}+{\;M}_{2}O_{3\;\left(s\right)}$$

BiFeO_3_ formation
(11)$$4{\mathrm{Bi}}_{\left(\mathrm{aq}\right)}^{3+}+6O_{\left(g\right)}\to 2{\mathrm{Bi}}_{2}O_{3\;\left(s\right)}$$
(12)$$2{\mathrm{Fe}}_{\left(\mathrm{aq}\right)}^{3+}+3O_{\left(g\right)}\to {\mathrm{Fe}}_{2}O_{3\;\left(s\right)}$$
(13)$${\mathrm{Bi}}_{2}O_{3\;\left(s\right)}+{\;\mathrm{Fe}}_{2}O_{3\;\left(s\right)}\to 2{\mathrm{BiFeO}}_{3\;\left(s\right)}$$

BiVO_4_ formation
(14)$$\mathrm{Bi}{\left({\mathrm{NO}}_{3}\right)}_{3\;\left(s\right)\;}+H_{2}O\to {\mathrm{BiONO}}_{3\;\left(\mathrm{aq}\right)}+2{\mathrm{HNO}}_{3\;\left(l\right)}$$
(15)$${\mathrm{NH}}_{4}{\mathrm{VO}}_{3\;\left(s\right)}+H_{2}O\to {\mathrm{HVO}}_{3\;\left(\mathrm{aq}\right)}+{\mathrm{NH}}_{4}{\mathrm{OH}}_{\left(l\right)}$$
(16)$${\mathrm{BiONO}}_{3\;\left(\mathrm{aq}\right)}+{\mathrm{HVO}}_{3\;\left(\mathrm{aq}\right)}\to {\mathrm{BiVO}}_{4\;\left(s\right)}+{\mathrm{HNO}}_{3\;\left(l\right)}$$

At first, dissolved bismuth nitrate and ammonium vanadate in water result in the formation of intermediate compounds of bismuth oxynitrate and vanadic acid, respectively. Under ultrasonic irradiation during the sonochemical process, the acoustic cavitation effect is induced, leading to rapid bubble growth followed by an implosive collapse of the bubbles. This collapse effectively generates a localized extreme hot spot that can provide high energy for a chemical reaction of two intermediate compounds to form bismuth vanadate particles.

When the composite is incorporated with Er dopant, its XRD pattern evidently exhibits the mixed phases of BiFeO_3_ and BiVO_4_ structures. The BiVO_4_ tetragonal structure exhibits peaks at 18.4° and 24.4°, which correspond to the crystallographic planes (101) and (200). The addition of Er dopant to the composite sample has a notable impact on the phase of the BiVO_4_ component. In the case of the bismuth vanadate/bismuth ferrite (BVO/BFO) composite, the BiVO_4_ component exhibits a monoclinic phase. When Er^3+^ is introduced into the composite, the Er^3+^ ion, with a smaller ionic radius (100 pm) than that of Bi^3+^ (117 pm), would prefer to substitute at Bi site, causing considerable distortion of the lattice structure of the BiVO_4_ crystal lattice and phase transformation from the monoclinic to the tetragonal phase. The cell parameters for all samples were calculated and are presented in Table 1, indicating the slight change in the lattice parameter values, which implies that the presence of the Er dopant in the composite samples did not significantly affect the lattice parameters [13,14].

### 3.2. Optical Property

The optical properties of all samples were assessed using the diffuse reflectance spectroscopic technique, and the corresponding reflectance spectra are shown in Figure 2. In the case of the pure BiVO_4_ sample, a distinctive absorption edge emerged at approximately 460–470 nm. This absorption edge was influenced by the pristine monoclinic phase in the crystallinity results and correlated with an optical band gap in the vicinity of 2.5 eV. For the as-prepared BiFeO_3_, the initial optical band gap was determined to be around 2.1 eV. After subjecting the sample to the calcination process at 500 °C for 2 h, the optical band gap increased to approximately 2.3 eV, which was in harmony with the crystalline result. Both composite samples and those containing various percentages of erbium displayed consistent patterns in their reflection spectra, with optical band gap values of around 2.2 and 2.8 eV. These values are associated with the tetragonal structure of BiFeO_3_ and tetragonal BiVO_4_, respectively, and this association was influenced by the phase formation of tetragonal BiVO_4_ in the rare earth element dopant specimens. The optical band gap calculations for all the samples were considered in determining the photoexcitation wavelength for the study of photocatalytic activity [15].

### 3.3. Chemical Bonding

The chemical bonding in all specimens was investigated using the Raman scattering technique, and the correlated results are represented in Figure 3. The Raman spectra of both pristine BiVO_4_ and BVO/BFO composite samples exhibit prominent scattering peaks at 816–818 cm^−1^, corresponding to the V-O symmetric vibration ($\nu_{s}$) mode. Additionally, doublet peaks associated with the symmetric–asymmetric vibration modes ($\delta_{s\;},\delta_{\mathrm{as}})$ of (${\mathrm{VO}}_{4}^{3-}$) group can be observed at 366 cm^−1^ and 326 cm^−1^, respectively [16,17]. These findings provide further support for the monoclinic structure of BiVO_4_. The emergence of a Raman peak at 850 cm^−1^ following the introduction of Er as a dopant in the BVO/BFO composite is indicative of a phase transformation from the monoclinic structure of BiVO_4_ to a tetragonal one. This observation confirms the beneficial effect of incorporating erbium into the BVO/BFO composite, leading to a transition towards the tetragonal phase. In addition, the Er-incorporated composite exhibited a strong luminescence band within the 400–750 cm^−1^ range when subjected to Raman laser excitation at 532 nm [18,19].

### 3.4. Surface Morphology

The surface morphology was observed using a transmission electron microscope (TEM), as depicted in Figure 4. In the case of the pristine BiFeO_3_ sample (Figure 4a), irregularly shaped particles with sizes in the range of 100 nm were observed. In Figure 4b, the BVO/BFO composite illustrates the appearance of small particles with indistinct shapes, with sizes around 200 nm. As shown in Figure 4c–e, the images of Er-doped BVO/BFO composites with varying percentages of Er loading exhibit small, square-shaped particles in the range of 100 nm. Furthermore, a higher percentage of Er doping led to the formation of smaller particles that clustered around larger particles, as well as the presence of small rod-like particles. Therefore, the influence of Er doping on particle formation resulted in a reduction in particle size and agglomeration of the powder sample due to the phase transformation of BiVO_4_’s monoclinic to tetragonal structure [20].

To confirm the phase formation of the composite sample, high-resolution transmission electron microscopy (HRTEM) images were taken, providing lattice d-spacing measurements for both the BVO/BFO composite and the 1% Er-doped BVO/BFO, as shown in Figure 4f,g, respectively. For the HRTEM image of the BVO/BFO composite, lattice d-spacing values of 3.1 Å and 2.5 Å can clearly be observed, corresponding to the (−112) plane of the monoclinic structure of BiVO_4_ and the (110) plane of the trigonal phase of BiFeO_3_, respectively. In contrast, the 1% Er-doped BVO/BFO sample reveals precise lattice d-spacing values of approximately 3.6 Å and 2.5 Å, corresponding to the (200) plane of tetragonal BiVO_4_ and the (110) plane of trigonal BiFeO_3_, respectively. This observation highlights the influence of the Er dopant on the BVO/BFO composite, affecting the phase transformation behavior of BiVO_4_ crystals. Specifically, the introduction of Er led to a transformation from a monoclinic to tetragonal phase within the BVO/BFO composite.

### 3.5. Surface Chemical Analysis

The chemical oxidation states observed through X-ray photoelectron spectroscopy (XPS) results serve to confirm the presence of all relevant elements in the Er-doped BVO/BFO composite sample. XPS was utilized for an in-depth investigation into the surface chemical oxidation state of the composite specimen, with the XPS spectra of the 1% Er-doped BVO/BFO composite displayed in Figure 5. Figure 5a reveals the deconvolution of Bi and Er oxidation state spectra. The binding energy of the Bi element can be observed at positions 159.90 eV and 165.25 eV, corresponding to the oxidation states of Bi 4f_7/2_ and Bi 4f_5/2_, respectively. Simultaneously, the oxidation state of the Er dopant can be prominently observed at peak positions 162.29 eV and 167.42 eV, aligning with the oxidation states of Er 4d_7/2_ and 4d_5/2_, respectively. For the observation of vanadium oxidation states presented in Figure 5b, binding energies at 581.1 eV and 520.1 eV indicate the oxidation states of V 2p_3/2_ and V 2p_1/2_, respectively. Furthermore, Figure 5c clearly reveals the Fe^3+^ component at a binding energy of 173.5 eV, along with a minor Fe^2+^ composition relocated at 710.1 eV and 715.9 eV for Fe^2+^ 2p_3/2_ and Fe^2+^ 2p_1/2_ states, respectively. The O1s species, observed at 531.5 eV, 532.9 eV, and 536.02 eV, was assigned to the oxygen vacancy on the surface (O_V_), lattice oxygen (O_L_), and adsorbed oxygen (O_A_) species for the BVO/BFO composite [21,22,23,24,25,26]. The observation of chemical oxidation states from the XPS results provides conclusive evidence for the existence of relevant elements and Er dopant in the BVO/BFO composite sample.

### 3.6. Piezo-Photocatalytic Performance

The observation of the decomposition of organic compounds was conducted to evaluate the photocatalytic, piezocatalytic, and piezo-photocatalytic performance under visible light and ultrasonic irradiation, as illustrated in Figure 6a–c and Table 2. The degradation of the compound was examined using various catalysts in different scenarios. The catalytic experiments were carried out via piezo-catalytic, photocatalytic, and piezo-photocatalytic activities under ultrasonic, visible-light, and combined exciting irradiation conditions, respectively. The results from piezo-catalytic performance testing exhibited superior efficiency, particularly in the case of the 3% Er-doped BVO/BFO composite sample. This sample achieved the decomposition of RhB under ultrasonic exposure alone with a decomposition rate of k = 0.0114 min^−1^, while the bare RhB organic dye exhibited only slight degradation under the same condition. The superior performance of piezo-photocatalysis was observed in the 1% Er-doped BVO/BFO composite sample, resulting in complete decomposition, with a decomposition rate of k = 0.0484 min^−1^ due to the higher specific surface area of 54.97 m^2^/g as illustrated in Figure 7a, which led to a larger active area for piezo-photocatalytic activity. This enhanced performance can be additionally attributed to the heterostructure of the catalyst, which inhibited the recombination of carrier generation in piezo-photocatalytic activity [27,28].

### 3.7. Scavenger Testing

The piezo-photocatalytic activity was assessed by conducting scavenger testing to investigate the active species, including hydroxyl radicals ($•{\mathrm{OH}}^{-}),$ superoxide radicals ($•O_{2}^{-}$), and hole radicals (h^+^), thus affecting the photocatalytic activity of the 1% Er-doped BiVO_4_/BiFeO_3_ sample. Ammonium oxalate (AO), p-Benzoquinone (BQ), and isopropyl alcohol (IPA) were employed as radical capping agents. The results of the scavenger trapping experiments are presented in Figure 6d. The lowest decomposition of RhB dye solution was observed in the photocatalytic reaction with BQ as the capping agent, while the higher decomposition of the RhB organic dye occurred when scavenger additives AO and IPA were used sequentially. These results suggest that the main active radical species are superoxide and hydroxyl radicals, as evidenced by the lower decomposition of the RhB organic compound observed when BQ and AO capping agents were added [29].

### 3.8. Impedance Property

Electrochemical impedance spectroscopy (EIS) was employed to investigate the charge transfer properties of the pristine BiVO_4_ (BVO) and 1% Er-doped BVO/BFO composite films under simulated solar irradiation (1200 W/m^2^). Figure 7b presents the Nyquist plots obtained from EIS measurements. The presentation of the charge transfer resistance results can be correlated with the photocatalytic performance of the material. The smaller diameter of the EIS arc observed for the 1% Er-doped BVO/BFO composite signifies a lower impedance compared to the bare BiVO_4_ sample. This directly translates to enhanced charge transfer efficiency within the Er-doped sample. These findings align well with the superior photocatalytic performance of the 1% Er-doped BVO/BFO composite in RhB degradation, as efficient charge separation is crucial for photocatalysis.

### 3.9. Piezo-Photocatalytic Mechanism

The piezo-photocatalysis mechanism can be elucidated through the energy band diagram theory of semiconductor heterojunction. Upon contact and application of excitation energy (light and ultrasonic sound), the stimulation of the BiFeO_3_ piezocatalyst under ultrasonic irradiation induced a piezo-potential, serving as the driving force to alter the band levels. This alteration provided the necessary energy for electrons/holes to engage in redox reactions. During ultrasonic irradiation, the direct excitation of electrons occurred through the collapse of bubbles in cavitation phenomena (ultrahigh pressure) on the catalyst interface. An indirect pathway involved the creation of electron–hole pairs from hot-spots (high temperature), generated by the collapse of bubbles in similar acoustic cavitation phenomena. The electrons generated were excited to the conduction band (CB), becoming free carriers, while holes were created in the valence band (VB) of the BiFeO_3_ catalyst [30,31]. Concurrently, the photocatalytic activity generated free electrons on the CB and hole carriers on the VB of the BiVO_4_ photocatalyst under visible light irradiation. At the interface of the piezo-photocatalyst composite, free electrons from piezocatalytic creation in the CB of BiFeO_3_ migrated to the nearby CB of the neighboring BiVO_4_ material. Then, they relocated to the surfaces of the catalysts, producing superoxide radicals $\left(•O_{2}^{-}\right)$ upon interaction with oxygen. In the case of the Er-doped BVO/BFO composite sample, the energy level of the VB slightly lowered due to the influence of the Er dopant contributing to the phase formation of the tetragonal phase of BiVO_4_, as indicated by the crystallinity results. The tetragonal phase exhibited a larger optical band gap than the monoclinic phase for the BiVO_4_ material. As illustrated in Figure 8, the higher level of the VB of BiFeO_3_ facilitated the easy movement of hole carriers from the VB of BiVO_4_ to the nearby higher VB of BiFeO_3_. In the subsequent steps, the separated carriers interacted with water or oxygen, producing hydroxyl $\left(•{\mathrm{OH}}^{-}\right)$ and superoxide ($•O_{2}^{-})$ radicals, respectively. These radicals play crucial roles in the decomposition of organic compounds. Consequently, the Er-doped BVO/BFO composite interface promoted charge carrier separation from piezo-photogeneration, leading to an enhanced organic compound decomposition performance [27,32,33,34]. In addition, the existence of the Er element in the composite could increase photon absorption in the visible region, leading to enhancement in electron–hole pair generation and the consequent photocatalytic performance of the composite.

## 4. Conclusions

In summary, sonochemical synthesis process was employed to synthesize Er-doped BiVO_4_/BiFeO_3_ composites. The XRD results indicate the existence of both BiVO_4_ and BiFeO_3_ as the major phases, with unobservable impurity phases. It is additionally implied that the incorporation of the Er dopant in BiVO_4_ significantly induced the phase transformation of BiVO_4_ from the monoclinic to tetragonal phase and resulted in a change in the size, shape, and morphology of BiVO_4_. The DRS results suggest that the samples possessed corresponding optical band gaps of two major phases in the composite. The superior catalytic performance was attained under both light and ultrasonic irradiation by the composite with a certain Er loading content of 1%, which could have been due to greater optical absorption by the Er dopant, a higher surface area, and the proper heterojunction between BiVO_4_ and BiFeO_3_ with assistance from piezo-induced charge separation, thus leading to an effective photocatalytic performance of the composite.

## Figures and Tables

**Figure 1 nanomaterials-14-00954-f001:**
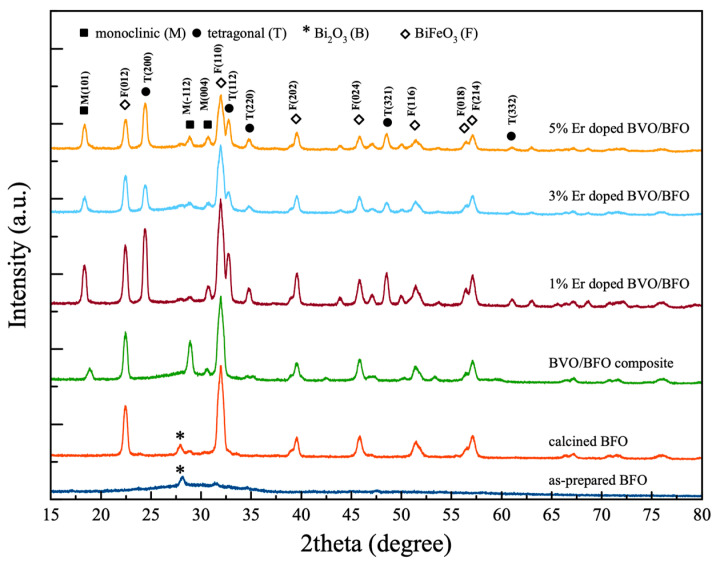
X-ray diffraction of pristine BiFeO_3_, BiVO_4_/BiFeO_3_, and Er-doped BiVO_4_/BiFeO_3_ composite samples.

**Figure 2 nanomaterials-14-00954-f002:**
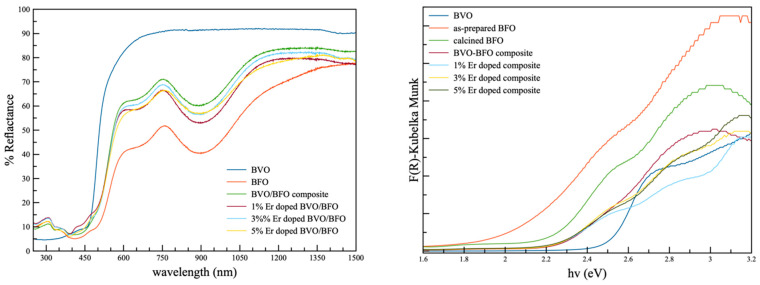
Diffuse reflectance spectrum and optical band gap calculation of pristine BiFeO_3_, BiVO_4_/BiFeO_3_, and Er-doped BiVO_4_/BiFeO_3_ composite samples.

**Figure 3 nanomaterials-14-00954-f003:**
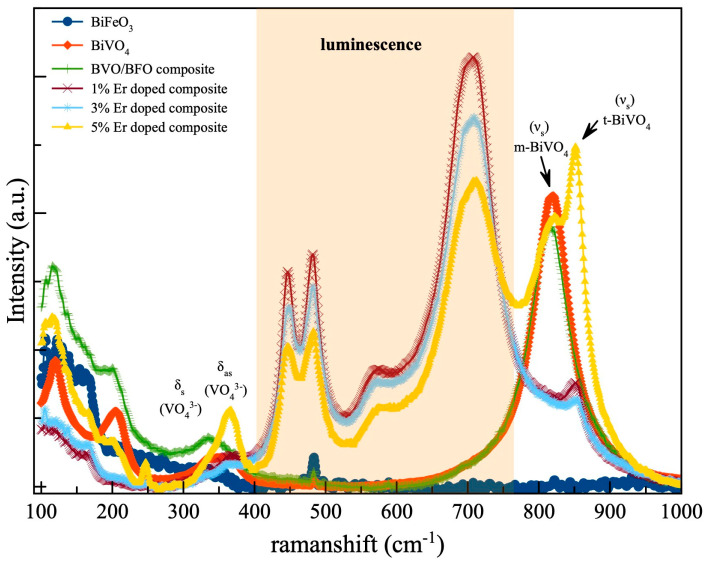
Raman spectra of BiVO_4_, BiFeO_3_, BiVO_4_/BiFeO_3_ composite and Er-doped BiVO_4_/BiFeO_3_ composite.

**Figure 4 nanomaterials-14-00954-f004:**
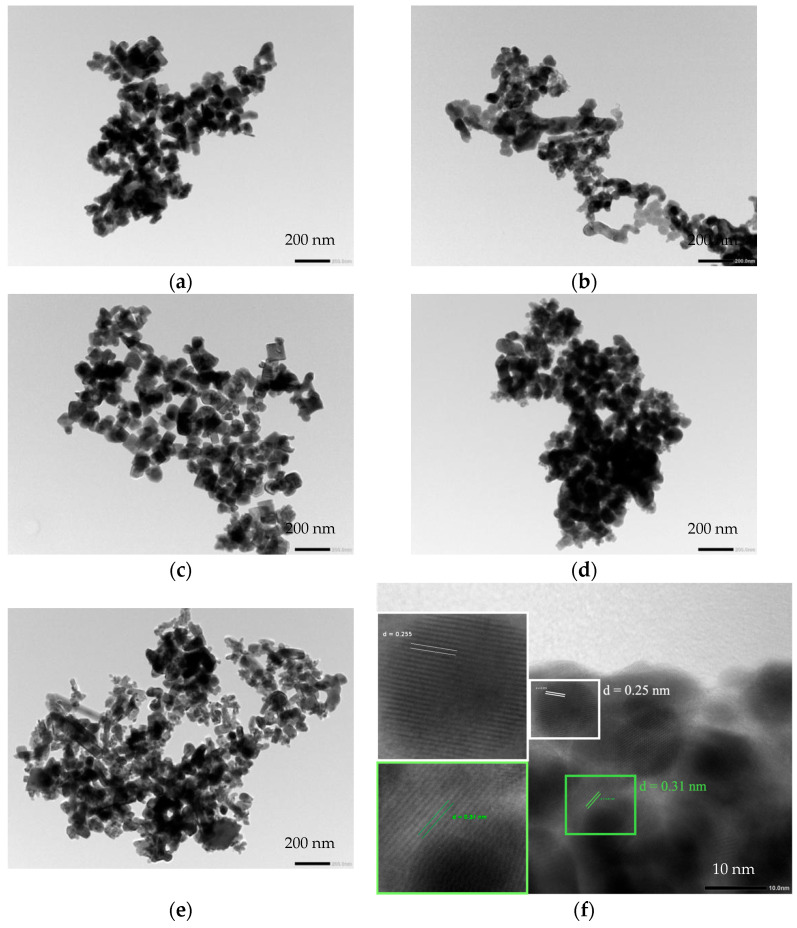
TEM images of (**a**) BiFeO_3_, (**b**) BiVO_4_/BiFeO_3_ composite, and (**c**–**e**) Er- doped composite sample varied at 1%, 3%, and 5% of Er incorporation, respectively. (**f**) HRTEM image of BiVO_4_/BiFeO_3_ composite. (**g**) HRTEM image of Er-doped BiVO_4_/BiFeO_3_ composite.

**Figure 5 nanomaterials-14-00954-f005:**
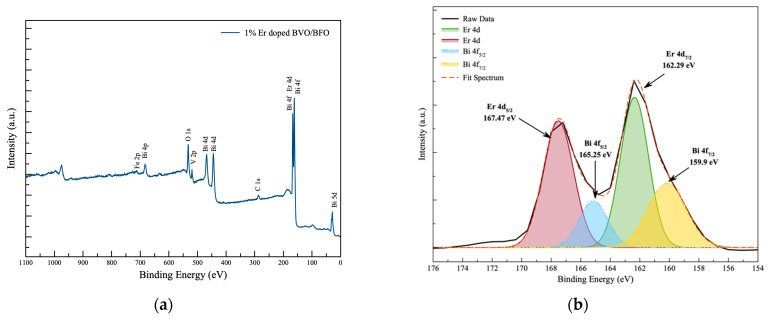
X-ray photoelectron spectroscopy spectra of (**a**) wide scan of 1%Er-doped BiVO_4_/BiFeO_3_; (**b**–**e**) deconvoluted fitting spectrum of Bi-4f, Er-4d, V-2p, Fe-2p, and O-1s spectra, respectively.

**Figure 6 nanomaterials-14-00954-f006:**
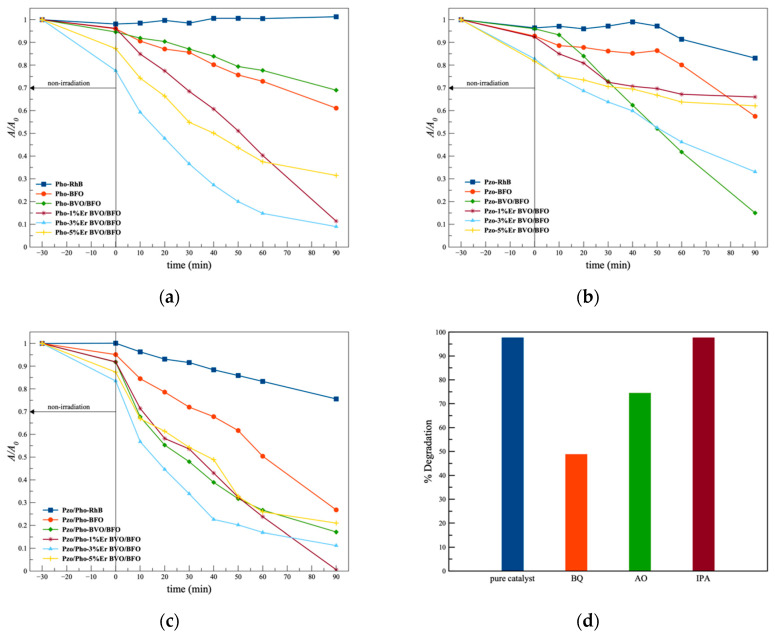
(**a**) Photocatalytic; (**b**) piezocatalytic; and (**c**) piezo-photocatalytic testing of all catalyst materials using Rhodamine B dye as organic compound model; and (**d**) scavenger testing for piezo-photocatalytic performance of the 1% Er doped BiVO_4_/BiFeO_3_ composite sample.

**Figure 7 nanomaterials-14-00954-f007:**
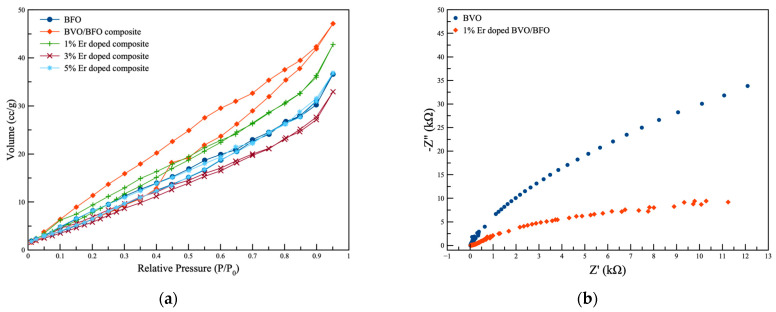
(**a**) Nitrogen absorption–desorption isotherm of all samples and (**b**) EIS plots of pristine BiVO_4_ (BVO) and 1% Er-doped BVO/BFO composite films under solar simulation light exposure (1200 W/m^2^).

**Figure 8 nanomaterials-14-00954-f008:**
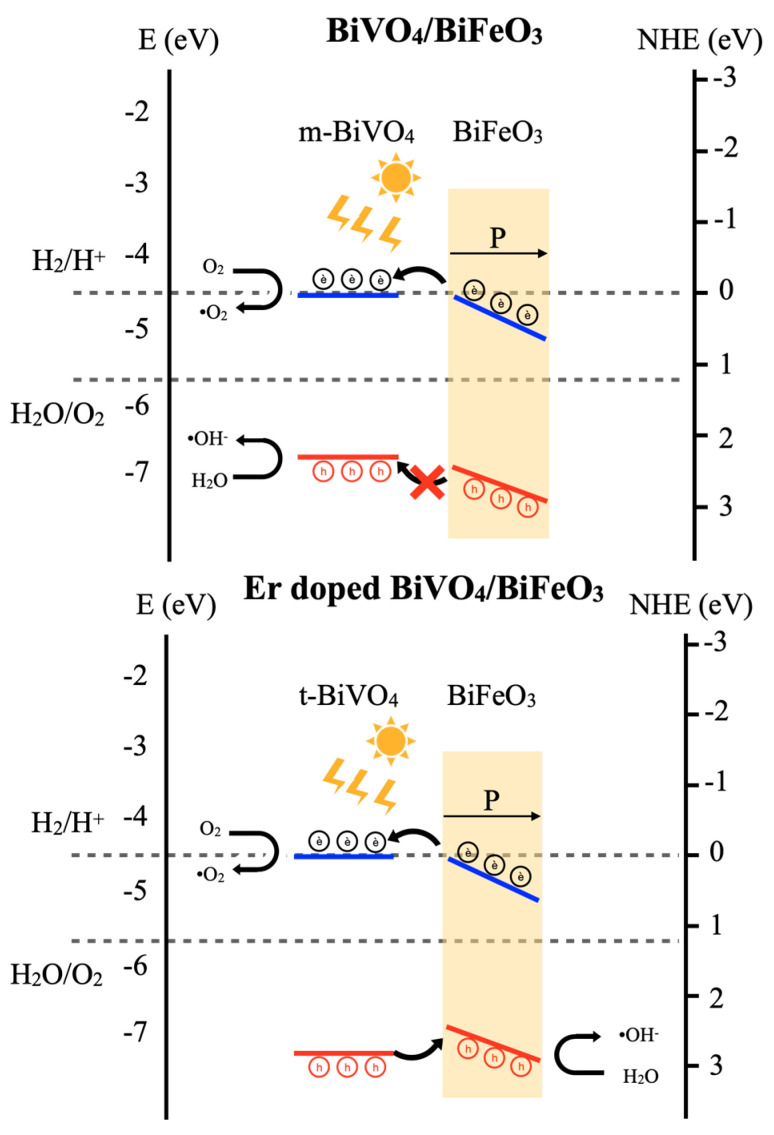
Piezo-photocatalytic mechanism of BiVO_4_/BiFeO_3_ and Er-doped BiVO_4_/BiFeO_3_ heterojunction under ultrasonic and visible light irradiation.

**Table 1 nanomaterials-14-00954-t001:** Calculated cell parameters of all samples.

Samples	Crystalline Size (nm)	Cell Parameters					
		a (Å)	b (Å)	c (Å)	α (°)	β (°)	Strain (ε)
BFO (trigonal)	16.05	3.12	3.12	3.12	156.76	156.76	1.403
BVO/BFO (trigonal)	13.81	3.11	3.11	3.11	157.26	157.26	0.085
1% Er-BVO/BFO (Tetragonal)	19.21	7.29	7.29	6.44	90.00	90.00	0.281
3% Er-BVO/BFO (Tetragonal)	15.83	7.29	7.29	6.44	90.00	90.00	1.983
5% Er-BVO/BFO (Tetragonal)	18.59	7.31	7.31	6.42	90.00	90.00	0.764

**Table 2 nanomaterials-14-00954-t002:** Specific surface area and decomposition rate of all samples.

Sample	Surface Area (m^2^/g)	Piezocatalyst		Photocatalyst		Piezo-Photocatalyst	
		*k* (min^−1^)	R^2^	*k* (min^−1^)	R^2^	*k* (min^−1^)	R^2^
BFO	38.41	0.0036	0.836	0.0049	0.997	0.0117	0.961
BVO/BFO composite	42.35	0.0160	0.924	0.0033	0.994	0.0201	0.993
1% Er-doped BVO/BFO	54.97	0.0049	0.934	0.0180	0.917	0.0484	0.824
3% Er-doped BVO/BFO	40.21	0.0114	0.980	0.0316	0.986	0.0332	0.963
5% Er-doped BVO/BFO	48.32	0.0037	0.986	0.0151	0.993	0.0202	0.977

## Data Availability

Data are contained within the article.

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
