# Peer review of "Er-Doped BiVO4/BiFeO3 Nanocomposites Synthesized via Sonochemical Process and Their Piezo-Photocatalytic Application"

_nanomaterials, 2024, doi:10.3390/nano14110954_

Round 1

Reviewer 1 Report

Comments and Suggestions for Authors

In this manuscript, the authors reported the preparation of Er-BiVO4/BFO nanocomposites, and investigated the piezo-photocatalytic performance toward RhB degradation. However, the paper lacks of novelty and interest. So referee suggests the paper is not suitable for publishing in Nanomaterials and should be rejected, because there are some major problems in the manuscript as follows.

1. In the background, what are the disadvantages of BiFeO3? Why choose the Er-BiVO4/BFO composite catalyst? The authors should present a detailed demonstration about this point.

2. Numerous errors about expression existed in this manuscript. For example, “doping with elements …to induce the reduction of host optical band gap…” Herein, the demonstration about doping is not correct. In fact, doping can also increase the optical band gap. Besides, in the preparation of BFO, the authors used “colloidal solution”, etc. The authors should pay more attention to accurate expression with well-chosen words. The authors should revise these incorrect expressions carefully and use standard terms.

3. According to the formula listed on page 3, BFO will be formed after experiencing the ultrasound treatment. However, the crystal phase of BFO could not be detected by XRD patterns. How to explain the disagreement?

4. Moreover, after experiencing the ultrasound treatment, Bi2O3 is formed by detected by XRD patterns. After calcination, Bi2O3 still existed in the product, what is the effect of Bi2O3 on the catalytic performance? What is the concentration of Bi2O3 in the product? Seemingly, the authors failed to consider this factor in discussing catalytic performance? This point is correlated with binary Er-doped BiVO4/BFO or ternary Er-doped BiVO4/ Bi2O3/ BFO? even correlated with the mechanism?

5. The picture should be clearer, with good visualization and high resolution, especially for HRTEM image.

6. For XPS analysis, why the intensity of Er 4d high-resolution XPS spectrum is much stronger than that of Bi 4f? The concentration of Er is only 1%, far below the content of Bi. How to explain this phenomenon? The authors should examine the data of the measurement carefully to accurately reflect the chemical status.

7. As for the discussion of catalytic performance toward RhB removal, the authors just presented the experimental results, and failed to in depth analyze the reason. For example, why the concentration of RhB increased upon exposure to visible-light irradiation? Why the removal rate of RhB over pure BFO is larger than that of BVO/BFO under photocatalysis? What is the difference of photo-excited carriers’ separation rate between pure BFO and BVO/BFO?

8. No sufficient results favored the mechanism of catalytic performance. The paper lacks of some important measurements, such as photocurrent, EIS plots, and Mott-Schottky plots and so on.  

Comments on the Quality of English Language

 English in the manuscript need to be improved.

Reviewer 2 Report

Comments and Suggestions for Authors

It could be explained  photocatalytic performance for its mechanism in detail. furthermore, it could be rewritten the introduction section.

Comments on the Quality of English Language

revise

Reviewer 3 Report

Comments and Suggestions for Authors

The manuscript first introduces Er-doped BiVO4/BiFeO3 composites prepared via sonochemical process with varying rare earth loading compositions. Remarkably, the Er-doped BiVO4/BiFeO3 composite heterojunction material demonstrates significant enhancement in piezo-photocatalytic activity under both ultrasonic and light irradiation, attributed to improved charge generation and separation. Furthermore, these promising findings highlight the strong influence of Er dopant on the phase transformation, morphology alteration, and optical band gap of the BiVO4 matrix. The experimental results and discussions are thorough and compelling. The manuscript’s thoroughness and depth make it suitable for publication in Nanomaterials. However, certain revisions are necessary for acceptance, as outlined below

1.    In the Introduction section, it is recommended to expand the background information and include more recent research findings, including references.

2.    Regarding Figure 4, the scale bars are unclear and the marked lattices in Figures 4f and 4g are indistinct. Kindly replace the enlarged version for clarity

Reviewer 4 Report

Comments and Suggestions for Authors The X-ray diffraction study presented in this manuscript seems insufficiently interpreted: the crystal cell parameters which depend on the synthesis conditions have not been determined and compared to standard crystallographic parameters. Identification from FCPDS files is insufficient to characterize the samples. A study of the state of crystallization would have been useful in order to statistically characterize the morphologies and microstructures of the polycrystalline phases thus synthesized. This would have been possible by a study of diffraction profiles using the Rietveld method for example. However, the other multiple experimental approaches are convincing. The main argument in favor of the role of heterojunctions in photocatalytic – piezocatalytic properties is provided by transmission electron microscopy studies. The TEM images show the proximity of BiVO4 and BiFeO3 crystallites and suggest the formation of possible heterojunctions BiVO4/BiFeO3 in the biphasic composites.

Round 2

Reviewer 1 Report

Comments and Suggestions for Authors

In this manuscript, the authors reported the preparation of Er-BiVO4/BFO nanocomposites, and investigated the piezo-photocatalytic performance toward RhB degradation. However, the paper lacks of novelty and interest. Detailed response about the revision was not found in the revised files. Referee suggests the paper should be rejected, because some representative problems were not rationally solved in the revised version as follows.

1. After calcination, Bi2O3 still existed in the product, what is the effect of Bi2O3 on the catalytic performance? What is the concentration of Bi2O3 in the product? Seemingly, the authors failed to consider this factor in discussing catalytic performance? This point is correlated with binary Er-doped BiVO4/BFO or ternary Er-doped BiVO4/ Bi2O3/ BFO? even correlated with the mechanism?

2. The picture should be clearer, with good visualization and high resolution, especially for HRTEM image. The authors should present HRTEM image with high magnification, not locally magnified area.

3. For XPS analysis, why the intensity of Er 4d high-resolution XPS spectrum is much stronger than that of Bi 4f? The concentration of Er is only 1%, far below the content of Bi. How to explain this phenomenon? The authors should examine the data of the measurement carefully to accurately reflect the chemical status.

4. As for the discussion of catalytic performance toward RhB removal, the authors just presented the experimental results, and failed to in depth analyze the reason. For example, why the concentration of RhB increased upon exposure to visible-light irradiation? Why the removal rate of RhB over pure BFO is larger than that of BVO/BFO under photocatalysis? What is the difference of photo-excited carriers’ separation rate between pure BFO and BVO/BFO?

5. No sufficient results favored the mechanism of catalytic performance. The paper lacks of some important measurements, such as photocurrent curves, and Mott-Schottky plots and so on.

6. Bi2O3 still existed in the product, however, in figure 8, the authors failed to consider the role of Bi2O3 in the composites for discussing the piezo-photocatalytic mechanism.